# Wearable Sensors, Data Processing, and Artificial Intelligence in Pregnancy Monitoring: A Review

**DOI:** 10.3390/s24196426

**Published:** 2024-10-04

**Authors:** Linkun Liu, Yujian Pu, Junzhe Fan, Yu Yan, Wenpeng Liu, Kailong Luo, Yiwen Wang, Guanlin Zhao, Tupei Chen, Poenar Daniel Puiu, Hui Huang

**Affiliations:** 1Singapore Institute of Manufacturing Technology, Agency for Science, Technology and Research (A*STAR), 5 Cleantech Loop, Singapore 636732, Singapore; 2School of Electrical and Electronic Engineering, Nanyang Technological University, 50 Nanyang Avenue, Singapore 639798, Singapore; 3Engineering Cluster, Singapore Institute of Technology, 10 Dover Drive, Singapore 138683, Singapore

**Keywords:** smart wearable, wearable sensor, pregnancy monitoring, artificial intelligence, data processing

## Abstract

Pregnancy monitoring is always essential for pregnant women and fetuses. According to the report of WHO (World Health Organization), there were an estimated 287,000 maternal deaths worldwide in 2020. Regular hospital check-ups, although well established, are a burden for pregnant women because of frequent travelling or hospitalization. Therefore, home-based, long-term, non-invasive health monitoring is one of the hot research areas. In recent years, with the development of wearable sensors and related data-processing technologies, pregnancy monitoring has become increasingly convenient. This article presents a review on recent research in wearable sensors, physiological data processing, and artificial intelligence (AI) for pregnancy monitoring. The wearable sensors mainly focus on physiological signals such as electrocardiogram (ECG), uterine contraction (UC), fetal movement (FM), and multimodal pregnancy-monitoring systems. The data processing involves data transmission, pre-processing, and application of threshold-based and AI-based algorithms. AI proves to be a powerful tool in early detection, smart diagnosis, and lifelong well-being in pregnancy monitoring. In this review, some improvements are proposed for future health monitoring of pregnant women. The rollout of smart wearables and the introduction of AI have shown remarkable potential in pregnancy monitoring despite some challenges in accuracy, data privacy, and user compliance.

## 1. Introduction

The health and well-being of pregnant women and fetuses are of paramount importance. The ability to comprehensively monitor fetal activity is critical for the early identification of potential complications and for ensuring optimal prenatal care. However, this is not always achievable. According to the WHO report “Trends in Maternal Mortality”, approximately 287,000 maternal deaths were recorded worldwide in 2020. In countries experiencing severe humanitarian crises, maternal mortality rates exceeded double the global average, reaching 551 maternal deaths per 100,000 live births, in contrast to the global rate of 223, and the global lifetime risk of death for pregnant women in 2020 was approximately 0.116% [1]. Conventionally, pregnancy care predominantly occurred within traditional healthcare settings, where women would attend routine check-ups and ultrasounds with their obstetrician or midwife [2]. These kinds of traditional methods for pregnancy monitoring often require frequent visits to medical facilities, which can be inconvenient and time-consuming for pregnant women. Therefore, remote and continuous monitoring is necessary. And it can be more challenging when it comes to some undeveloped areas where medical support is not enough. However, with the emergence of healthcare information systems, pregnancy care has transcended the confines of conventional healthcare settings. Now, women can receive care remotely thanks to health apps, wearables, telemedicine, and remote medical consultation systems. These advancements are reshaping pregnancy care, offering women personalized, real-time health information and insights into the well-being of their unborn children. Additionally, healthcare professionals are equipped with richer data to effectively manage pregnancies [3].

Multimodal wearable monitoring systems for pregnant women have been a popular research field in recent years among all digital health technologies. Compared to traditional monitoring methods, there are at least five functions that make wearable devices suitable for healthcare: wireless mobility, interactivity, sustainability, miniaturization, and wear resistance [4]. Typically, a complete multimodal pregnancy-monitoring system consists of multiple sensors, one or more cloud servers, and interaction between patients and mobile devices, as well as patients and medical staff, as shown in Figure 1. Different sensors are applied to obtain related physiological signals, and common wear positions on a pregnant woman’s body are provided. The physiological signals are then transmitted to the cloud server via a cellular network or directly examined by clinicians. Beyond traditional interaction between medical staff and patients, clinicians can store diagnostic results in the cloud server and check the feedback and recorded data. Through mobile devices (e.g., mobile phones), recorded data and diagnostic results in cloud servers can be accessed at any time by users. This transmission method ensures that pregnant women can easily view their historical physiological and diagnostic records. With the smart wearables, the health parameters of the mother and fetus, such as electrocardiogram (ECG), maternal heart rate (MHR), fetal heart rate (FHR), and fetal movement (FM), can be continuously collected so that potential complications can be detected in the early stage.

The common types of pregnancy-monitoring wearable sensors are shown in Figure 2. They are classified into five categories based on the measurement parameters and mechanisms. Bio-potential sensors are based on changes in electrical potential, aiming at directly measurable electrical signals, e.g., ECG. Pressure and inertial sensors are based on physical movement, mainly targeting the movement of the mother and the fetus during pregnancy, e.g., FM. Acoustic sensors are based on the acquisition and separation of sound, mainly measuring regular acoustic signals, e.g., FHR. Electrodermal sensors, a relatively new research direction, analyze the composition of liquids to confirm physiological parameters. They are mainly suitable for measuring stress and emotional states, and their use in pregnancy monitoring needs to be further explored. All of the above sensors can be integrated into multimodal smart wearable devices. The acquired signals are then transmitted and analyzed to get the result. AI algorithms revolutionize the traditional approach to pregnancy monitoring by offering real-time analysis of vital signs, FM, and maternal health indicators. These sophisticated algorithms provide early detection of potential complications, such as preeclampsia, allowing for timely interventions to safeguard both maternal and fetal health [5,6]. With the convenience of remote monitoring through wearable devices and mobile apps, AI empowers pregnant women with continuous access to personalized care, reducing the need for frequent clinic visits [7]. This application also facilitates the collection and management of different physiological states of pregnant women, providing them with personalized medical advice and treatment plans. With a wide range of sensors and algorithms emerging, a clear understanding of wearable sensors, data-processing algorithms, and applications of AI in maternal health monitoring is essential.

However, the further spread of wearables and the adoption of AI face multiple challenges. First, many AI algorithms require large amounts of computational resources, yet wearable devices are often limited by battery capacity and power management, which restricts the real-time processing capability of complex AI models. Second, the personal physiological data collected by wearable devices are very sensitive, and how to ensure user privacy and security during data transmission and storage is an important challenge. Third, the physiological characteristics of different users vary widely, making it difficult for a single AI model to adapt to all users, and the development and deployment of personalized models are difficult. Finally, wearable devices often require real-time monitoring and feedback, but the complexity of AI algorithms may lead to computational delays, affecting user experience.

This paper is structured as follows. Section 1 provides a brief introduction to smart pregnancy monitoring, including basic concepts, functions, components, and workflow. The classification of common wearable sensors is also introduced. Section 2 presents the review methodology of literature screening. Section 3 focuses on wearable sensors for maternal and fetal health monitoring. Then, Section 4 explores multimodal monitoring and sensor integration for comprehensive health assessments. Section 5 presents the commonly used data-processing flows and analysis of different algorithms, as well as the application and impact of AI in relevant areas. Finally, Section 6 provides the conclusion and outlook.

## 2. Review Methodology

The review protocol adhered to the guidelines in the Preferred Reporting Items for Systematic Review and Meta-analysis Protocols (PRISMA-P) [8]. The objective of this review is to assess the advancements in research about wearable sensors for monitoring maternal and fetal health during pregnancy, as well as data-processing techniques and potential applications of AI technologies in this domain.

### Literature Selection Strategy

The literature search encompassed the following databases: Google Scholar, IEEE Xplore, Science Direct, MDPI, ACM Digital Library, and Scopus. Table 1 shows the inclusion and exclusion strategy of this review. Initially, we used free-text search terms such as “wearable sensor OR wearable”, “maternal OR pregnant”, “health monitoring”, and “(artificial AND intelligence) OR AI”. A stricter strategy was implemented for a detailed analysis of the selected papers. To ensure that the focused articles were both academic and up to date, we focused on the articles that are after 2000 and peer-reviewed. Then, we excluded all articles that were not written in English. We also noted certain articles that did not include wearable sensors in their content, were not aimed at monitoring the health of pregnant women, or used a questionnaire format as the primary research method. Based on the scope of this paper, we excluded these articles. Finally, we excluded articles whose content was excessively antiquated and could be replaced by more recent results.

As shown in Figure 3, after removing duplicates, non-English, and antiquated literature, a total of 197 articles were identified from databases. Among them, 91 articles were selected based on title and abstract screening. Finally, 34 articles were selected to be eligible.

## 3. Wearable Sensors for Maternal and Fetal Health Monitoring

### 3.1. Bio-Potential Sensor

Bio-potential sensors are pivotal devices that bridge biology and electronics, enabling the detection and analysis of various analytes with high precision. These sensors include a wide array of types, notably ECG, electroencephalogram (EEG), and electrohysterogram (EHG), which will be the primary focus of this discussion. Central to these sensors is the conversion of biological interactions into electrical signals, a process facilitated by electrodes and biological-recognition elements like enzymes or antibodies immobilized on their surface [9,10]. The use of nanomaterials enhances electrode performance, increasing sensitivity due to their large surface areas and efficient electron-transfer capabilities. Advances in bioelectronics, especially through miniaturization, further improve these sensors’ sensitivity and reduce their invasiveness. The stability of the immobilized biological components and the transduction efficiency are crucial for the biosensor’s accuracy, making bio-potential sensors essential tools in diagnostics, environmental monitoring, and beyond [11,12,13].

#### 3.1.1. ECG

ECG is a common medical test used to measure the electrical activity of the heart over a period. A traditional ECG system is operated by placing electrodes on the patient’s skin, which detects the tiny electrical changes on the skin that arise from the heart muscle’s electrophysiological pattern during each heartbeat. Recently, wearable technologies like the Apple Watch have integrated ECG functions, allowing users to monitor their heart rhythms continuously with a small device worn on the wrist. These wearable ECG devices provide the convenience of tracking heart health in real-time and in everyday settings. Despite these advancements, challenges remain in maintaining signal quality, managing power consumption, and ensuring long-term reliability, especially in more dynamic and complex non-clinical environments.

In the field of wearable technology for pregnancy health monitoring, the significance of ECG monitoring has been highlighted through various innovative approaches to sensor technology, emphasizing the potential for enhanced comfort, accuracy, and non-invasiveness in healthcare tracking.

The study by Galli et al. [14] is an excellent exemplar for illustrating how a basic fetal ECG (fECG) system operates. As shown in Figure 4, this novel FHR monitoring system used four dry, textile electrodes, which were developed to overcome the limitations of traditional gel-based systems. The setup featured four electrodes strategically placed around the mother’s belly to maximize signal-to-noise ratio (SNR), with an additional reference electrode positioned on the hip. This configuration was crucial in enhancing signal quality and reliability. The textile electrodes, due to their material properties, were prone to triboelectricity artifacts—unwanted electrical charges generated by friction, which could affect signal clarity. To overcome this, the system employed a hybrid (both hardware and Matlab 2019b software) pre-processing step to mitigate these artifacts and improve the signal’s fidelity for further analysis. This system was tested with both semi-simulated and real data. It showed improved performance metrics such as sensitivity and root mean-square error over traditional methods, but fully eliminating these interferences remains a challenge, particularly during extended monitoring periods.

The integration of liquid metal electrodes for ECG monitoring, as introduced in the study [15] through the development of a Kirigami-structured Liquid Metal Paper (KLP), also provided a promising approach in the field of fECG. This new type of wearable ECG sensor featured a stretchable, ultrathin, and recyclable electrode system that was self-supporting and exposed the conductor. It significantly enhanced the ability to capture high-quality electrophysiological signals, marking a significant improvement in both design and functionality. However, such materials are still in the early stages of development, and challenges remain regarding their durability and the cost-effectiveness of large-scale production. Another study by Yang et al. [16] further contributed to this field by presenting a flexible, wearable, and wireless fECG monitoring system. They created a system using flexible electrode patches and a special mathematical method that could precisely track FHR and UC. This system also provided a comfortable way for long-term health monitoring outside the hospital.

In 2019, another study by Balakrishna et al. [17] further explored the complexities of wearable health monitors, focusing on a fECG monitor as a case study. The paper explored the design challenges of ensuring precise signal acquisition through sensor design, data processing, and communication. Furthermore, the experiments of integrating printed sensors into IoT garments were also presented. The research highlighted the complexity of sensor design and the need for innovative solutions to address the use of wearable devices in remote monitoring and healthcare management. In the same year, Gao et al. [18] explored the potential of flexible electronics in wearable sensing. They suggested that material innovation and sensor design were crucial for effectively monitoring changes in health status. Two years later, Li et al. [19] innovated a multifunctional epidermal sensor by developing a nanocomposite hydrogel that integrated MXene (Ti_3_C_2_T_x_), polyacrylic acid (PAA), and amorphous calcium carbonate (ACC). This study contributed significantly to the field of wearable technology by crafting a sensor that combines flexibility, rapid response (within 20 ms), and exceptional self-healing properties with environmental degradability. These features made it highly suitable for applications such as electronic skin, health monitoring, and the enhancement of human–machine interactions. The sensor’s advanced functionality underscored its potential in personal healthcare diagnostics and offered a new avenue for developing sustainable, user-friendly wearable technologies.

The study by Sutha and Jayanthi [20] highlighted advancements in bio-potential sensors for fECG monitoring, focusing on enhancing power efficiency and signal accuracy. These developments are important for creating more portable and reliable fetal health-monitoring systems, but further validation is needed to ensure long-term performance in diverse clinical settings. Similarly, non-invasive techniques combining ECG and phonocardiography [21,22,23] present promising alternatives to traditional monitoring methods, although issues related to signal interference and real-time accuracy require additional research.

Furthermore, a review by Hong et al. [24] on wearable and implantable devices for cardiovascular healthcare emphasized the crucial role of flexible and stretchable electronics in continuous cardiac monitoring, which was directly applicable to advancements in maternal and fetal ECG monitoring. However, many wearable sensors face durability issues when exposed to real-world conditions over prolonged periods, particularly in non-clinical environments. Lastly, a systematic review by Alim et al. [25] and a survey by Wakefield et al. [26] reflected on the current state and public perception of wearable sensors for pregnancy health monitoring. These studies underlined the importance of further validation and widespread adoption of these technologies into routine medical practice, alongside addressing cost, usability, and data privacy concerns.

These developments collectively underscore the dynamic evolution of ECG monitoring technologies, promising a future where maternal and fetal health monitoring is more accessible, accurate, and non-invasive. Nonetheless, challenges such as ensuring long-term durability, maintaining signal accuracy in dynamic environments, and achieving cost-effective large-scale production remain significant hurdles to widespread adoption.

#### 3.1.2. EEG

EEG is a crucial technique in neurophysiological studies, facilitating the non-invasive recording of electrical activity within the brain. By measuring voltage fluctuations resulting from ionic current flows within neurons, EEG provides valuable insights into brain function and has been widely used in both clinical and research settings. Initially employed for adult brain studies to diagnose conditions like epilepsy, sleep disorders, and brain injuries, its applications have included monitoring fetal brain activity.

Recent advancements in fetal EEG (fEEG) technology aim to overcome previous limitations, such as signal artifacts and interference from maternal movements. Historically, fEEG faced significant challenges due to the difficulty of obtaining clear signals from the fetal brain amidst the noise generated by maternal tissues and movements. Innovations in electrode design, including the development of more sensitive and less invasive electrodes, have been pivotal in improving signal quality.

The study in [27], a systematic review to assess the feasibility and clinical utility of fEEG monitoring during labor, identified fEEG patterns associated with different labor conditions, including fetal distress and drug effects. The review indicated recent improvements, which included automated algorithms for signal analysis of fEEG to reduce interpretation bias. In addition, the review emphasized the need for further clinical studies to validate fEEG monitoring in real-world settings, as well as for continuing innovation in electrode design and signal processing to enhance reliability.

Another work by Lew et al. [23] developed a new non-invasive scanner to monitor the electrical activity of the uterus, fetal heart, and fetal brain. The study was simulated with a model of a pregnant torso containing a 35-week-old fetus, using an array of magnetic field sensors as well as passive and active shielding to reduce interferences. The results indicated that the scanner could reduce outside interference by 1000 times, making it easier to detect inside signals from the body. It could also visualize uterine activity and estimate currents around the uterus. The study highlighted the need for practical testing of the scanner. While it presented an innovative approach to non-invasive monitoring, it relied on simulations and did not address the cost or practicality of using such a scanner in hospitals. The study also demonstrated the use of a full-coverage fetal-maternal scanner for non-invasive monitoring of electrophysiological activity, indicating its potential in pregnancy healthcare. The initial FM scanner design is represented in Figure 5, conceptualizing how these sensor arrays might be utilized in practice, with optically pumped magnetometer (OPM) sensors integrated into a lightweight format. The FM scanner was an innovative step in biometric monitoring, offering a glimpse into the potential for future non-invasive medical diagnostics. These advancements underlined the growing importance of non-invasive technologies in prenatal care, offering promising pathways for early detection and intervention in fetal-distress scenarios.

#### 3.1.3. EHG

EHG is a non-invasive technique used to monitor uterine electrical activity. Unlike traditional methods, such as intrauterine pressure catheter (IUPC) and tocodynamometer (TOCO), which can be invasive and affected by operator variability, EHG provides a safer alternative for assessing UC. By detecting electrical signals generated by uterine muscle contractions, EHG offers a promising approach for long-term and real-time monitoring in clinical settings. Recent advancements in EHG technology have significantly improved its accuracy and clinical applicability.

The study in [28] compared the efficacy of an electrical uterine monitor (EUM), TOCO, and IUPC in labor, demonstrating the potential of EUM for accurate UC assessment. The results supported the superiority of non-invasive EMG technology over external tocodynamometry. This was complemented in the work of Wang et al. [29], which explored electro myometrial imaging (EMMI) to assess UC with greater spatial resolution, suggesting EMMI’s clinical utility for three-dimensional uterine electrical activation mapping.

In another study, Hao et al. [30] epitomized the progressive utilization of EHG signals in UC monitoring by integrating AI with EHG sensor data. As shown in Figure 6, the research adeptly developed a convolutional neural network (CNN) model to discern UC from EHG signals, employing a robust dataset comprising 122 recordings from an Icelandic 16-electrode EHG database (DB1). The CNN was trained using five-fold cross-validation, demonstrating high performance with metrics such as sensitivity (0.87), specificity (0.98), accuracy (0.93), and AUC (0.92). The model was further validated on an independent dataset (DB2), where it also maintained strong generalization with comparable results. Compared to traditional signal-processing techniques, the use of AI in this context not only reduced manual interpretation time but also significantly improved the overall accuracy and reliability of UC detection. This fusion of AI and EHG technology represents a critical step toward the application of AI in clinical pregnancy monitoring, enabling real-time, large-scale data processing with high reliability, even in complex and dynamic monitoring environments.

Moreover, the study in [31] highlighted the acquisition and analysis of EHG signals as an effective alternative to traditional methods for monitoring FHR and UC, pointing towards the reliability of EHG in reflecting the UC. Furthermore, a wearable patch device was developed for EHG monitoring by Jo et al. [32], showcasing the device’s ability to predict preterm birth risk, indicating its practicality for continuous pregnancy health monitoring.

These studies collectively underscored the evolution of EHG-monitoring technologies from basic signal acquisition to sophisticated imaging and wearable devices, marking significant strides toward enhancing prenatal care through improved detection and prediction of labor-related complications.

### 3.2. Inertial Sensor and Pressure Sensor

Inertial sensors and pressure sensors have become integral to advancements in health-monitoring technologies, particularly in the field of pregnancy monitoring. Inertial sensors, including accelerometers and gyroscopes, measure acceleration and rotational motion, respectively. They work by detecting changes in velocity and orientation, providing crucial data on body movements and postures. These sensors are widely used to monitor FM, maternal physical activity, and UC, offering valuable insights into both fetal well-being and maternal health.

Pressure sensors, on the other hand, detect variations in pressure, converting them into electrical signals. These sensors are highly sensitive and flexible, capable of measuring minute pressure fluctuations with high accuracy. They are mainly employed to monitor UC by detecting the pressure exerted on the abdominal wall, thereby providing real-time data on contraction patterns and intensities, which are crucial in assessing labor progress and identifying potential complications such as preterm labor.

Recent developments in this technology have focused on enhancing sensor sensitivity, flexibility, and integration into wearable devices. Some of the sensors are inspired by the nuanced and highly sensitive mechanoreceptors of human skin and have evolved to become essential components in wearable technology, providing detailed insights into health and movement with minimal intrusion, particularly for monitoring pregnancy-related parameters.

Lei et al. [33] developed a self-healing, mechanically flexible ionic skin sensor using a bioinspired mineral hydrogel for the sense of minor pressure fluctuations. This sensor represented the move towards designing devices that replicate human skin’s multifunctionality and mechanical adaptability for precise pressure detection, which could be highly applicable in continuous pregnancy monitoring. Likewise, Xiong et al. [34] introduced a capacitive pressure sensor, using convex microarrays, characterized by its flexibility and extreme sensitivity. This innovation highlighted the sensor’s ability to track physiological signals and robotic hand movements, illustrating its applicability in health surveillance and human–robot interactions, as well as its potential for monitoring UC in pregnancy.

Another typical work by Lee et al. [35] contributed to this field by developing a conductive fiber-based ultrasensitive textile pressure sensor for e-textiles, as shown in Figure 7. The integration between fabric and sensor, without compromising sensitivity or flexibility, paved the way for its application in wearable technology, such as systems for monitoring health. Trung et al. [36] reviewed the progress in flexible and stretchable physical sensors based on contractive fiber, highlighting their role in personal healthcare and wearable human-activity monitoring. Their work emphasized the significance of these sensors in creating sensor-integrated platforms for real-time health status monitoring, which also presents a promising direction for pregnancy-monitoring applications.

In 2018, a skin-inspired hierarchical polymer architecture was proposed for triboelectric sensors, achieving high sensitivity without the use of spacers [37]. This design enhanced the sensor’s ability to detect minute pressures and movements, simulating the efficiency of human tactile sensing. Additionally, the use of liquid metals in stretchable and soft electronics was explored, offering a perspective on creating highly deformable yet conductive materials for wearable electronics [38]. Such materials may offer new avenues for creating sensors capable of tracking maternal and fetal health metrics during pregnancy.

The role of these sensors in pregnancy health monitoring is indispensable. Inertial sensors track body movements and postures, offering critical information about physical activity levels and FM. Pressure sensors enable accurate monitoring of UC, which is vital for assessing labor progress and identifying potential complications. Recent innovations, such as self-healing ionic skin sensors and conductive fiber-based textile sensors, have further enhanced the functionality of these devices and their integration with wearable health-monitoring systems. These advancements promise minimally invasive, continuous health tracking, enabling early detection of complications and improving overall healthcare in pregnancy.

### 3.3. Electrodermal Activity Sensor

The integration of electrodermal activity (EDA) sensors into wearable technologies has significantly advanced the monitoring of stress and emotional states, which is particularly important in the health of pregnant women. EDA sensors measure the electrical conductance of the skin, which varies with the moisture level due to sweat gland activity. This activity is directly influenced by the autonomic nervous system, making EDA a reliable indicator of physiological arousal and emotional states, which are critical in monitoring maternal stress during pregnancy.

EDA sensors work by applying a small, imperceptible electrical current through the skin and measuring the resulting conductance. When a person experiences stress or heightened emotional states, sweat gland activity increases, leading to higher skin conductance levels. These sensors are thus capable of capturing subtle physiological changes associated with stress and emotional arousal, which can help monitor the mental and emotional health of pregnant women in real time.

In the study of Jang et al. [39], the team introduced a novel wearable sensor system using graphene e-tattoos (GET) for unobtrusive and high-fidelity EDA sensing on the palm. This system aimed to address the limitations of existing EDA sensors, such as obstructiveness and poor signal fidelity. As shown in Figure 8, researchers developed a stretchable GET system connected to a rigid EDA wristband using heterogeneous serpentine ribbons (HSPR). This design minimized strain at the interface between flexible and rigid components. Validation through finite element modeling and experimental tests showed that the GET system provided high-fidelity EDA signals with minimal motion artifacts, outperforming traditional gel electrodes. Results demonstrated that this GET system reduced strain concentration significantly and maintained robust performance during various daily activities over a 15 h monitoring period. The system showed strong correlations with gold-standard EDA sensors, confirming its reliability and making it suitable for continuous monitoring during pregnancy.

In the study by Sun et al. [40], the team reviewed the rise of metal–organic frameworks (MOFs) and their potential applications in e-skin technologies and artificial intelligence. MOFs were highlighted for their customizable porosity, electrical properties, and high sensitivity, making them ideal candidates for next-generation sensors. The study emphasized the advantages of using MOFs in flexible and stretchable electronics, particularly for health-monitoring applications. In the context of EDA sensors, MOFs have the potential to significantly improve sensor performance by enhancing signal accuracy and reducing motion artifacts. Sun et al. developed a conductive MOF material that demonstrated improved sensitivity and signal fidelity in wearable applications, offering promising avenues for real-time emotional and stress monitoring in pregnant women. Such technologies hold great potential for improving maternal health monitoring by accurately tracking stress levels.

In the study of Poh et al. [41], a novel wrist-worn sensor was developed for unobtrusive, long-term EDA assessment outside the laboratory, marking a significant step toward continuous monitoring of emotional states in real-world settings. This technology held promise in tracking the mental states of expectant mothers, offering insights into their stress levels and emotional well-being throughout pregnancy. Another study in [42] reviewed the challenges and opportunities presented by wearable EDA sensors, emphasizing the importance of artifact handling and the development of skin-conformal electrodes for improving measurement quality. These considerations are crucial for ensuring accurate and reliable emotional-state monitoring of pregnant women.

In 2019, Affanni [43] highlighted the design of a dual-channel EDA and ECG sensor for measuring mental stress from hands, addressing motion artifacts, and enhancing the versatility of EDA measurements. This approach could enhance the assessment of pregnant women’s stress levels by providing a comprehensive overview of their physiological responses. In the investigation reported in [43], construction workers’ perceived risk was evaluated through the EDA sensor, showing its capability to differentiate between low- and high-risk activities through physiological responses. This methodology could similarly be adapted to study stress and emotional reactions in pregnant women during various activities, improving pregnancy monitoring.

Song et al. [44] introduced a simplified on-skin-printed sensor modality for simultaneous mechanical and bioelectrical sensing, showing the potential for multifunctional wearable sensors in health monitoring. This innovation could be applied to develop more comprehensive wearable devices for pregnant women, capable of monitoring both physiological pressure and emotional states.

By employing psychophysiological sensors to evaluate mental workload during web navigation, Jimenez–Molina et al. [45] highlighted the sensors’ ability to measure mental states in interactive tasks. This approach could inform the development of interventions aimed at reducing stress and improving emotional well-being in pregnant women.

Lastly, the reliability of wearable EDA sensors over traditional research-grade equipment was validated in two typical studies [46,47], confirming their effectiveness in capturing accurate physiological data. This validation supports the integration of EDA sensors into wearable technologies for comprehensive health monitoring, marking a significant step towards advanced healthcare through continuous emotional and stress monitoring.

In the context of pregnancy healthcare, EDA sensors provide valuable data on the psychological well-being of pregnant women. By continuously monitoring stress levels, these sensors help healthcare providers identify periods of high stress, which can negatively impact both the mother and the developing fetus. This information is crucial for developing interventions to manage stress and improve overall maternal health. Additionally, the non-invasive nature of EDA sensors makes them suitable for long-term monitoring, providing continuous data that can inform both clinical practices and research studies. As a result, EDA sensors are becoming an essential tool in the comprehensive monitoring of pregnancy health, contributing to better health outcomes through early detection and management of stress.

Therefore, the application of EDA sensors in wearable technologies presents a promising avenue for enhancing pregnancy health monitoring. The advancements in sensor design, data-processing techniques, and wearable integration provide a foundation for future research and development endeavors aimed at bolstering the emotional and psychological well-being of pregnant women.

### 3.4. Acoustic Sensor

Acoustic sensors, especially those based on fetal phonocardiography (fPCG), have emerged as a pivotal technology in non-invasive pregnancy health monitoring. The mechanism detects the acoustic vibrations produced by the fetal heart, offering a passive, low-cost method for assessing fetal well-being. The advancements in this domain range from the development of novel sensor designs to the enhancement of signal processing techniques, aimed at improving the precision and dependability of FHR tracking.

Charlier et al. [48] introduced a significant innovation with AcCorps, an affordable, 3D-printed acoustic sensor tailored for fPCG. This device aimed to address the limitations of conventional acoustic sensors by optimizing acoustic amplification in the low-frequency band, a critical factor for capturing fetal heart sounds. Its potential for accuracy improvement in FHR monitoring of this design was corroborated through in silico testing, test bench applications, and trials involving pregnant volunteers.

Khandoker et al. [49] developed an affordable, non-invasive system for fPCG that captures the sounds of the fetal heart using four piezoelectric sensors as shown in Figure 9. It showed a high correlation (r = 0.96) with fECG measurements, providing a promising alternative to traditional fetal-monitoring methods. The study emphasized the potential of fPCG in improving prenatal care by offering a reliable and simple solution for monitoring FHR, particularly in low-resource settings.

Furthermore, the integration of acoustic sensors with other technologies has been explored to broaden the scope of pregnancy health monitoring [48]. For instance, Qin et al. [50] developed a wearable system to detect FM, incorporating a three-axis acceleration sensor alongside the acoustic-sensing elements. This system represented a comprehensive approach to FM monitoring, aiming to surpass the constraints of conventional approaches by offering a continuous, precise evaluation.

Aravindan et al. [51] further exemplified the application of fPCG in remote monitoring devices. Their work focused on the development of a wireless wearable device with an acoustic sensor for fPCG, alongside electro hysterography for monitoring UC. This device aimed at improving access to obstetric care, particularly in rural areas, by enabling remote, real-time monitoring of pregnancy health indicators.

These developments underscore the evolving landscape of sensor technology, specifically for pregnancy monitoring. By leveraging innovations in sensor design and integration, researchers are paving the way for more accessible, accurate, and comprehensive healthcare solutions. These advancements promise to enhance the monitoring of maternal and fetal well-being, enabling better detection of potential health issues and improving overall healthcare outcomes.

## 4. Multimodal Sensor Integration

The integration of multimodal sensors into wearable devices represents a significant leap forward in maternal and fetal health monitoring. These advancements not only streamline the simultaneous collection of various physiological signals but also enhance the accuracy and reliability of health assessments during pregnancy.

The study in Ref. [52] presented an innovative monitoring platform that integrated advanced flexible electronics and wireless technology. This platform was designed to improve maternal and fetal health by providing comprehensive, continuous, and non-invasive monitoring of vital signs across various healthcare settings. The system utilized three flexible, soft sensors that monitor vital parameters for both mothers and fetuses, with capabilities such as cuffless blood pressure measurement, EHG for uterine activity assessment, and automated body position classification. The outstanding effectiveness and adaptability of this system were shown in both high-resource environments, such as Chicago, and low-resource settings, such as Lusaka and Zambia, demonstrating its potential to reduce maternal and neonatal morbidity and mortality by facilitating early detection and intervention for pregnancy-related complications.

Another important study by Chung et al. [53] delineated a pioneering approach in the realm of neonatal and pediatric care through the deployment of skin-interfaced biosensors for wireless physiological monitoring as shown in Figure 10. This innovative technology circumvented the limitations of traditional tethered systems, enabling non-invasive, continuous tracking of vital signs like heart rate, respiration rate, temperature, and blood oxygenation. The biosensors’ integration with Bluetooth technology facilitated real-time data transmission, enhancing patient mobility and caregiver interaction. Notably, the system extended its capabilities beyond conventional monitoring, encompassing movement tracking, kangaroo care assessment, and acoustic analysis of cardiac and vocal biomarkers, thus offering a holistic view of the patient’s physiological state. Clinical validation in pilot studies underscores the system’s accuracy and reliability, corroborating its potential to revolutionize neonatal and pediatric critical care by fostering a more conducive, less intrusive monitoring environment.

A recent study by Du et al. [54] introduced a wearable device that could assess the relative position, force, and duration of FM using multi-point Inertial Measurement Unit (IMU) sensing coupled with real-time classification techniques. This integration allowed for comprehensive monitoring compared to traditional single-parameter devices. Such a system, validated through both phantom simulations and clinical tests, demonstrated high accuracy and correlation in recognizing FM and was well-received by pregnant women.

Adding to the multimodal approach, the study in [55] proposed a multi-parameter wearable system equipped with fiber Bragg grating sensors that are sensitive to strain and could monitor the cardiorespiratory parameters and detect FM at the same time. This single sensing unit enclosed within a polymeric mask improved sensitivity and comfort, marking a step toward integrated health monitoring. The interconnection of wearable sensors with telemedicine, as discussed by Kalasin and Surareungchai [56], provided a framework for remote health monitoring by gathering real-time physiological and biochemical information. The challenge remained to develop sensors capable of long-term, continuous health monitoring at the molecular level, a critical aspect of advancing telemedicine. Maugeri et al. [57] emphasized how wearable sensors can aid in studying fetal and pregnancy outcomes. This comprehensive overview underscored the potential of wearables in collecting extensive data that could lead to better health predictions and interventions.

Moreover, advancements in fetal position detection using different techniques showed how sensor integration could contribute to more accurate assessments of fetal well-being and development [58]. Source localization from multisensory magnetocardiographic recordings had also been employed to track FM, demonstrating the feasibility of integrating data from various sensors to provide a dynamic picture of fetal health [59]. The wearable sensor glove, as developed in [60], exemplified a practical digital health application combining electrodermal activity and pulse-wave analysis sensors into a single, user-friendly device. This glove underscored the practicality and potential of wearable technology in healthcare. The study in [61] also systematically reviewed wearable sensors for pregnancy monitoring, highlighting the diverse applications and the importance of integrating various sensing modalities for a holistic health-monitoring approach.

Lastly, a wireless, remote solution for FHR and MHR monitoring at home allowed for non-invasive, continuous monitoring, reinforcing the growing trend towards remote healthcare solutions [25,62].

## 5. Data Processing

The process of obtaining reasonable detection results from raw digital data typically involves several steps, as depicted in Figure 11. Data analysis in the server is represented in the dotted line field. After obtaining the raw data from wearable sensors, valid signals are screened out by pre-processing. Then, the signals are analyzed and classified using threshold-based or AI-based algorithms depending on different conditions. Afterward, physiological signals with distinctive features are acquired. Finally, diagnostic results are generated from the data. In the majority of situations where useful real-time conclusions or diagnoses through long-term monitoring are needed, data-analysis capabilities need to be equipped with wearable sensors or their upper-level processors to facilitate pregnant women and reduce the burden on medical personnel. This section summarizes the algorithms and development for data processing in recent years and outlines the impact and application of AI.

Table 2 lists several representative research classified with different acquired signals and data-processing algorithms.

### 5.1. Data-Transmission Methods

The raw data acquired by the wearable sensors need to be transmitted to upper levels, including analysis equipment and cloud servers. The current data-transmission methods mainly include Bluetooth [65,67,69,70,71], wired protocol [65,68], WIFI [70], NB-IoT [72], LoRa [73,74] and some equipment specialized in data acquisition [64,66]. Table 3 selects some typical recent literature that lists performance comparisons between different communication protocols, including power consumption, security, latency, etc. Of these, Bluetooth remains the most mature and common technology, while emerging, IoT-friendly technologies such as LoRa and NB-IoT are gradually becoming more popular.

Bluetooth is one of the most common methods used in current research for data transmission. For example, in the study [69], the analog-to-digital converter (ADC) at the proximal end packaged the sampled signals into frames and transmitted them to a mobile phone for processing via a low-power Bluetooth module. The device could work continuously for 15 h. Another example was study [71], in which the signal transmission module could transmit the ECG data via Bluetooth to a PC or store the data directly on a memory card.

WIFI was primarily used for transmission from mobile devices to cloud servers. The study [70] was a combination of Bluetooth and WIFI. The data were first transmitted to the mobile device via Bluetooth, where it had undergone pre-processing. Subsequently, the pre-processed data were securely and wirelessly transmitted via WIFI to the cloud and all signals were processed then. The data could be downloaded in real time by the pregnant woman and healthcare team. Wired protocols were also preferred in certain situations like Doppler ultrasound. In the study [65,68], signals could be output directly through the device ports. It was notable that this method of data transmission was not suitable for home-based and long-term monitoring. In addition, commercially available data-acquisition equipment like PowerLab has already been developed to handle multiple types of data [64,75]. In the study [66], there was a requirement to collect a variety of signals, including acoustic, acceleration, force, inertia, and maternal sensation. The researcher designed the signal conditioning circuits and built two DAQ systems independently, which were placed on both sides of the abdomen to acquire and record signals from eight sensors.

NB-IoT is a cellular network-based low-power wide-area protocol designed for remote medical-monitoring devices that require ongoing data transmission. Its primary advantage is very low energy consumption, making it well-suited for wearable sensor applications. And the security is based on SIM encryption and authentication for cellular networks. However, its limited bandwidth poses a challenge, making it less suitable for platforms with complex multimodal sensors [76]. A study [72] suggests a hybrid solution combining NB-IoT and Wi-Fi. The aim of the research is to assess the power consumption of the proposed hybrid system in comparison to single-radio NB-IoT technology. A test board was created, and multiple data-transmission experiments were conducted using both NB-IoT and Wi-Fi radios. The findings indicate that a hybrid approach can enhance system efficiency while maintaining acceptable transmission range times. The study also explores how transmission frequency impacts battery life, revealing that more frequent transmissions lead to shorter device lifespan and higher costs.

LoRa technology is a wireless communication method designed for long-range, low-power data transmission. It supports extended-distance communication at lower data rates, making it ideal for IoT devices that require wide-area coverage, minimal power usage, and limited data transmission. Unlike NB-IoT, LoRa operates on unlicensed frequency bands, which are more cost-effective but come with higher risks. The security is based on AES-128. Additionally, LoRa often requires the installation of private gateways and servers [76]. Study [74] outlines a LoRa-based system integrated with smart textiles, incorporating sensors for photoplethysmography, electrocardiograms, and body temperature monitoring. The researchers also developed healthcare applications compatible with LoRa-enabled smart textiles. The article primarily evaluates signal strength over various distances, with results indicating an effective outdoor communication range of up to 250 m at 915 MHz, 350 m at 433 MHz, and up to 50 m indoors.

In recent years, smart wearables have been on the rise, and researchers have explored some faster and more resource-efficient ways of analyzing data. Due to the computing and storage capacity development of wearable sensors, the analysis of physiological data acquired in real time can be performed with the assistance of fog computing [77]. The research established a health-monitoring framework based on IoT and fog computing for obtaining physiological parameters such as temperature, blood pressure, and ECG to analyze the health status of pregnant women. Acquired physiological data were first transmitted to the fog node for preliminary analysis. Valid data were then further analyzed by the cloud server. This approach avoided the latency associated with cloud computing and made it possible to quickly obtain real-time analysis in the acquisition of large volumes of data.

From the above study, smart wearable devices have high requirements for data transfer. Electronic health records (EHR) have been iteratively updated over the years and are becoming increasingly popular among patients and physicians. EHRs provide a convenient way to share and manage relevant data between patients and healthcare providers. But real-time data transfer and storage become major challenges in integrating EHR and wearable sensor data.

### 5.2. Pre-Processing

Signal pre-processing aims to prepare valid signals for data analysis. Typically, signals from wearable sensors are one-dimensional time domain data, such as ECG, HR, respiration, and acceleration. This step usually consists of data validation, segmentation, denoising, and detrending.

Data validation is necessary before the analysis. It is primarily about removing the invalid signal segment to ensure the effectiveness of subsequent steps. Wearable devices usually have some data processors integrated on them. However, to consider portability, the performance of the device is usually limited. Data validation serves the function of reducing the amount of data transferred, compressing the data, and optimizing the energy consumption in some architectures. In this case, we need to consider quality of service (QoS). QoS is a set of techniques used to manage network resources and ensure that network traffic is delivered efficiently. It enables you to prioritize different types of traffic so that important traffic passes through first and helps to prevent network congestion. A systematic review of the latest research on QoS optimization in smart healthcare applications is presented in study [78] and demonstrates the use of ML, cloud computing, and IoT in smart healthcare. The survey shows that many QoS metrics focus on how to reduce total energy consumption and increase network throughput while extending the network lifecycle and ensuring more reliable communications by reducing congestion and conflicts.

Segmentation is formulated according to the needs of different data-analysis algorithms, which may be based on training and test sets, or a certain length to facilitate subsequent analysis. After that, the data need to be standardized to ensure that they conform to the structured data formats required by EHR systems. It is worth noting that wearable sensor data formats and communication protocols are often not standardized, with different device manufacturers using different data models and standards. So, interoperability of different standards is also a major challenge.

The focus lies on denoising and detrending, which are more complicated and have a greater impact on performance. There has been a great deal of mature work on denoising one-dimensional signals in existing research. Several efficient and simple methods have been developed, such as FIR [62,64,69,70], IIR [66,79,80,81], Kalman filter [63,67], CWT and DWT [65,68,69], and Hampel [65]. In practical research, it is common to use a mixture of methods to achieve the best performance.

The FIR filter is commonly used due to its simplicity and speed. A certain type of FIR can eliminate random noise, and it is also known as a moving average or moving median filter. FIR can also target and attenuate noise in a certain frequency range and is effective when there is an apparent difference between the noise and the valid signal. A typical example is the study [62], where a high-pass filter, a low-pass filter, and a powerline filter were successively used to obtain the FHR and MHR signal. In the study, the acoustic signal and ECG were acquired and processed separately using a combination of filters. A moving average filter was first used for the ambient noise. A Butterworth filter with a cut-off frequency of 85 Hz was then used to get the useful signal frequency intervals. And powerline filters with suitable frequencies were used to optimize the samples collected from different countries. Finally, an additional moving median filter was used to deal with low-frequency noise. The system of the article uses a bi-linear parallel mode to process two signals at the same time. And an app form is used as the UI of the system, and the database is deployed directly on the mobile phone.

IIR filters also have a place in practical applications because of their ease of design and higher filtering accuracy. In the study in [70], which was focused on UC signals, a DC-blocking filter was first used to remove the biopotential signal, followed by an IIR notch filter to deal with the power line noise. An IIR filter with five cut-off frequencies ranging from 10 to 30 Hz was then used to improve the detectability of UC by expanding the search for maternal phonocardiography (PCG) signals to encompass a wider range of characteristics. All the physiological and movement signals in the article are encapsulated into data packets and transmitted to the mobile device via Bluetooth. The data are then transferred wirelessly and securely from the mobile device to the cloud application via WiFi. The signals are processed at the cloud server level. In another example [81], which focused on the FHR filter, the IIR filter was also used in pre-filtering and selection. A zero-phase IIR filter was used to pre-process the seismo-cardiogram (SCG) signals. The research showed the filtered wavelet and compared it with other sensors. After combining the information from three sensors, further algorithms could be used to process the valid signals.

For cases where multiple sensor signals need to be analyzed, the time axis of the sensors may be misaligned. The CWT, on the other hand, can convert the signal into the time-frequency domain, allowing the fusion of desired frequency components while retaining time-domain variations. Its application was shown in the analysis in [81]. Pregnant women’s SCG and gyro-cardiogram (GCG) were obtained to record the FHR. The vibration amplitude components of the different sensors were not aligned, so a direct unification operation of the amplitudes would be misleading. But in recent years, CWT has been gradually replaced by DWT, which is more suitable for wearable sensors because of its greater advantages in memory and computing speed. The ability to pre-process data at the sensor level makes it suitable for long-term monitoring as well.

The study in [69] used DWT to obtain the correct contraction signal. It divided the original signal into multiple layers and obtained the approximate and detailed components. The detailed coefficients of each layer were set according to the general frequency of contractions. Then, the problem of baseline wandering was solved using the limit detection and correction algorithm from the study in [82]. Similar important works addressing baseline wandering were also presented in [83,84,85]. They may have future applications in smart pregnancy monitoring based on wearable sensors.

The study in [68] showed that DWT could also be applied for FHR monitoring. In the study, data from a Doppler ultrasound system were analyzed, and 85 different mother wavelets were evaluated and screened. Two datasets including clinic and public were used to assess the validity and accuracy of DWT. The result showed an accuracy of higher than 95% and suggested that DWT could be applied for similar remote fetal-monitoring methods.

In practice, sensor signals are easily contaminated by noise, and Kalman filtering is an effective preprocessing method to recover the signal. An example was presented in [63], which focused on the processing of contaminated FM signals. The study developed a new model using the Kalman-filtering algorithm, time, frequency, and wavelet domain (TFWD) feature-extraction methods, and the Bayesian optimization algorithm to identify FM.

After the valid signals have been filtered, individual outliers need to be excluded. The most widely used tool is “Hampel”, a function that calculates the median of a window of samples, including several surrounding samples and their standard deviation about the median of each sample’s window. For instance, in the study in [65], if a sample differed from the median by more than three standard deviations, the median would replace it.

### 5.3. Data-Analysis Algorithms

After pre-processing, data analysis is applied to reveal the hidden, researcher-interested parts. Over the past few years, algorithms for data analysis of wearable sensors can be broadly divided into two categories: threshold-based algorithms and AI-based algorithms.

#### 5.3.1. Threshold-Based Algorithms

According to the working principles, threshold-based algorithms can be divided into peak point detection [62] and local maximum detection [68,79,80]. This type of algorithm is mainly used when the signal is simple, with low noise, or when denoising is effective.

The principle of the threshold-based algorithm is very simple. In physiological signal analysis, abnormal signals tend to have lower or higher values than healthy signals. Based on physiological and statistical analysis, we can define one or more thresholds to mark events of interest. This enables the classification of physiological signals by simply determining the range of values. Threshold-based algorithms usually perform poorly in practical applications. The reason is that such algorithms rely on fixed thresholds, which are susceptible to individual differences and environmental changes and may lead to false alarms or omissions. For example, the intensity of FM may vary depending on the size of the pregnant woman or the position of the fetus, and a single threshold may not be suitable for all situations. In recent years, several adaptive algorithms with dynamic thresholds have been developed to partially ameliorate this problem.

In the study in [66], a threshold-based approach was used to detect FM with a fusion of signals from different numbers of sensors. Signals from multiple sensors including acoustic sensors, pressure sensors, and accelerometers were used. After signal fusion, the areas where the extremes occurred densely were found as the suspected time of FM, and the overlap was taken as the result of the detection. The results showed that when analyzing each sensor individually, the algorithm could detect 97% of the FM, but 66% of the detections were false positives, which reduced the performance of the system. When fusing the signals from multiple sensors, the precision improved to 86% and the accuracy decreased moderately to 85%. The threshold-based algorithm suffered from the lack of accuracy in preprocessing signals from wearable sensors, but performance could be improved by additional operations such as the fusion analysis of multiple sensors. The DAQ system is equipped with an on-board micro-SD card for data storage and synchronizes pregnant women’s sensing and signal acquisition via handheld buttons.

Another good example is a study on FHR [68]. The study searched for the local maximum in a segment of Doppler ultrasound signal to screen the waveforms of the heartbeat, followed by deriving the FHR. Then, the cross-correlation of signals and Bland–Altman statistical analysis were performed. Compared with the clinic data, the validity of the algorithm could be 95.03%. However, it is essential to note that the algorithm here is based on reliable results during data pre-processing.

Threshold-based methods have also been used for long-term health monitoring. In the study in [86], a non-wearing algorithm was used to analyze the sedentary time of maternal women and identify the time when they were not wearing the sensors. The data obtained by the algorithm were compared with the tester’s daily self-report, the final Bland–Altman plot revealed no bias, and the mean absolute percent errors were less than 10%. In the study, monitoring over multiple time lengths was conducted. It was found that increasing the threshold of the algorithm could effectively reduce the possibility of misclassification in long-term detection as compared to short-term detection.

#### 5.3.2. AI-Based Algorithms

In previous research, pregnant women were examined in professional medical environments, so threshold methods were widely used due to their simplicity. However, wearable sensors are usually used in diverse environments and require considerable accuracy at the same time. Moreover, due to the rapid advancement in machine-learning research and hardware performance, threshold-based algorithms have been gradually replaced by AI-based methods (also known as machine-learning methods), which are more accurate and applicable in the field of health monitoring by wearable sensors. Some of the AI-based algorithms that have been widely used in recent years are the linear model [64,65,66,67,69,87], neural network [64,66,69], decision tree [63,64,65], SVM [64,65,66,69,70], ensemble learning model [63,64,66,69,87], clustering model [65,67], and statistical model [63,64].

The performance of a machine-learning algorithm is presented in terms of the confusion matrix, as shown in Figure 12. Categorical indicators, such as $Accuracy=\frac{TP+TN}{TP+TN+FP+FN}$, $Precision=\frac{TP}{TP+FP}$, and $Recall=\frac{TP}{TP+FN}$, can be obtained from the confusion matrix.

The F1 score, as shown in Equation (1), is a common measure for classification problems, which is particularly useful when dealing with imbalanced datasets or when both precision and recall are equally important. It is commonly used in tasks like medical diagnosis. It is the harmonic mean of precision and recall, ranging from a maximum of 1 and a minimum of 0. A higher F1 score means that the algorithm is more effective.
(1)$$F1=\frac{2\times Precision\times Recall}{Precision+Recall}$$

The study in [88] used a linguistic-based decision-analysis tool to detect early onset of preeclampsia by detecting pregnant women for risk of hypertension. The fuzzy language was applied to the approach in two stages. Firstly, the raw signals were linguistically transformed to make the data interpretable and flexible, followed by knowledge extraction and categorization of the data using decision trees. However, high interpretability and high accuracy could not be achieved at the same time for the method used in the study. It was prone to overfitting or low accuracy.

Another typical example of using decision trees for conception-seeking women’s health monitoring is the study in [87]. In the study, the feasibility of wearable bracelets in detecting parameters such as wrist temperature and heart rate was explored. And an algorithm capable of recognizing fertility windows in real time was implemented using decision trees. The algorithm could detect a 6-day fertility window in 90% of menstrual cycles, enabling higher accuracy and more precise targeting.

The study in [63] used a modified decision tree model, LightGBM, to detect FM. The features in TFWD were extracted to train a LightGBM classifier, which was an effective implementation of the Gradient-Boosting Decision Tree (GBDT). To evaluate the hyperparameters of LightGBM, the Grid Search algorithm (GSA), Random Search algorithm (RSA), and Bayesian Optimization algorithm (BOA) were used in the study. The optimal hyperparameter values under different optimization algorithms were obtained. Due to its lowest loss in cross-validation, BOA was chosen to evaluate the promising hyperparameter values. Eventually, the accuracy and F1 score of FM detection could reach 94.06% and 96.85%, respectively.

The studies in [64,65,66] all targeted FM signals and used a variety of AI methods for comparison. In the study reported in [66], the researchers compared four common AI-based classifiers. The results showed that the neural network-based classifier was the best (F1 = 0.79, accuracy = 0.9), followed by the support-vector machines (F1 = 0.78) and random forests (F1 = 0.77), and the logistic regression classifier was the worst (F1 = 0.73). Remarkably, the threshold-based classifier was also implemented in the study, but its overall performance was not as good as the AI-based one, even when compared to a logistic regression classifier.

The study in [65] used the Synthetic Minority Oversampling Technique (SMOTE) function to increase the number of minority samples, enhancing the balance between the two types of data before training, which may have an unpredictable effect. The study used nine commonly used AI-based methods to classify FM data. The model performance was evaluated using cross-validation. The F1 score of the Extra Trees Classifier (ETC) is above 0.8. This suggested that it could potentially replace the current widely used ultrasound methods to supply more information about the fetus under long-term monitoring at home or in other environments. The raw acceleration data of the whole system is transferred from the accelerometer to the system chip and then to the smartphone.

Different AI-based algorithms were also compared in an earlier study [64]. In the study, the FM data measured by accelerometers were collected. Then the processed data were classified and evaluated by different algorithms. The results indicated that the Bagging classifier was the most effective, and the random forest classifier was the fastest in terms of calculation. The study also analyzed other factors that may influence the performance. For example, the analysis of artifacts was the focus of FM monitoring because of the unavoidable maternal body movement. It was concluded that as the artifact concentration increased, the algorithm accuracy decreased. Feature sample size was also important in AI-based algorithms. The study investigated the variation in accuracy when different numbers of extracted features were used. The results revealed that as the number of features increased, the accuracy showed a roughly logarithmic increase, and after a certain number of features, the addition of new features did not improve the accuracy significantly.

### 5.4. Data Privacy and Management

Patient physiological data collected from wearable sensors involve highly sensitive personal information in EHR. Therefore, how to ensure the security and suitable management of these data becomes an important issue. This part introduces the typical data privacy challenges and potential solutions, as well as how to properly manage private data.

First, during data transmission, most wearable sensors use the techniques introduced in Section 5.1 to transmit data to gateways or cloud platforms. The use of end-to-end encryption is necessary to prevent data from being stolen or tampered with during transmission. Also, whether the data are stored locally by the wearable device or transmitted to the server or cloud, the data should be protected using encryption when stored. Secondly, anonymization and de-identification can be used. Data need to be anonymized before they are transmitted to the cloud or analytics platform, or random identifiers must be generated to replace the actual identity information. This can ensure data integrity and availability without revealing personal information. Third, during data processing, in order to reduce the risk of data exposure, wearable devices can make use of edge computing technology to conduct part of the data processing locally. Unnecessary data transmission and storage can be reduced, as well as potential privacy risks. Finally, it needs to be ensured that patients authorize data collection in an informed manner through clear and understandable privacy policies and data usage instructions [89,90].

There are already some more mature security protocols in place to address such issues, such as HIPAA (US Health Insurance Portability and Accountability Act) and GDPR (EU General Data Protection Regulation) [91,92]. Future development of wearable sensors must focus on data security to prevent sensitive maternal health data from being compromised.

Wearable sensor platforms typically use database technologies that are different from traditional databases in processing large amounts of real-time data. The databases used by these platforms need to have the ability to handle massive, high-speed, and continuous data while meeting the requirements of low latency, high availability, and scalability. Compared with traditional relational databases, these new database technologies have greater flexibility and concurrent processing capabilities and can better cope with the complex data requirements brought by wearable devices.

### 5.5. AI in Pregnancy Monitoring

Over the past decades, AI has been increasingly used in new disciplines, including long-term maternal and fetal health monitoring. And it is far more than a simple classification issue. Today’s AI is different from a single algorithm in that it is a group of algorithm combinations that can improve and create its own logic in response to new data. As data-storage technology matures and the amount of data continues to grow, it has become possible to collect large amounts of relevant data on the internet to train AI. Furthermore, the increasing demand for health monitoring has prompted scientists to fuse AI with wearable sensors, enabling smart healthcare for pregnancy.

Pregnancy monitoring with AI generally includes three aspects: early detection, smart diagnosis, and lifelong well-being. Early detection is essential for pregnant women to avoid the serious results of certain illnesses like preeclampsia, as well as to help determine important points in the pregnancy, such as labor. For example, in the study reported in [69], a lightweight AI algorithm was developed and deployed on resource-constrained IoT devices. The researchers of the study detected UC to determine preterm labor onset. Physiological data were received and uploaded to an edge server, where it was analyzed by AI to remind pregnant women to prepare for labor, and uncertain data were uploaded to a cloud server for diagnosis with more powerful models. Early detection is more mature, as in most cases the AI only needs to determine whether it is “normal” or not and does not need to provide further information.

Smart diagnosis is more challenging and needs exploration for pregnant women. Enabling AI to make medical diagnoses like a human doctor requires a large amount of data and powerful arithmetic to train models. In fact, Google’s DeepMind Health has been exploring smart diagnosis for quite some time [93]. DeepMind cooperated with the Royal Free London NHS Foundation Trust to develop a powerful and accurate diagnosis system based on a large number of datasets. However, there are still several challenges that need to be addressed, such as public trust, personal data security, and the risk of AI error. Building public trust in novel areas is a difficult and lengthy process. Personal data security is also an important consideration in the context of the big data era, and the relevant laws need to be improved. The risk of AI error is a further complicating factor, as the rules for allocating responsibility remain a blank slate.

Telemedicine systems combined with AI can bring lifelong well-being to patients including pregnant women, and AI has partially replaced human customer service to answer questions and provide care. Literature [94] surveyed the acceptance of health chatbots by internet users and concluded that health chatbots were acceptable to most respondents, although hesitancy towards AI technology could affect users’ engagement. This suggested that there is great scope and potential for AI technology to be used in lifelong well-being. The research reported in [95] developed an AI-based mobile application that chats with pregnant women, offering comprehensive healthcare information from pregnancy through postpartum.

However, there are some challenges to applying AI in healthcare. First, the performance of AI in medical diagnosis depends on the dataset used for model training. If the training data are biased or insufficient, the AI model may perform poorly in specific groups. For example, AI may diagnose common diseases better but may perform poorly on rare conditions, making them susceptible to misdiagnosis. Second, many AI algorithms, especially deep-learning models, have decision-making processes that are not transparent to users and medical professionals. Third, there is the issue of data privacy. AI models require a large amount of medical data for training, including patients’ medical records, imaging data, and even genetic information. If the data are not managed properly or the storage system is hacked, it could trigger a privacy breach. In practice, there are several possible solutions to the risks of AI diagnosis. Firstly, AI should not be relied upon exclusively but should be used as a tool for the doctor and not as a substitute for decision-making. Second, “interpretable AI” is used in diagnosis or with accompanying information such as confidence levels and suspected causes to assist the doctor’s judgment. Third, a multi-layer protection mechanism is gradually established; for example, after the AI provides a diagnosis, the doctor must review it.

The potential of combining AI with wearable sensors in healthcare is huge and is increasingly becoming a part of the healthcare system for pregnant women. There will also be many innovative technologies and problems emerging in future research waiting to be explored and solved.

## 6. Conclusions and Outlook

This review provides a comprehensive review of wearable sensors, pregnancy health monitoring, data processing, and the impact of artificial intelligence (AI).

In recent years, wearable sensors for pregnancy monitoring have made significant advancements, offering continuous, non-invasive, real-time tracking of maternal and fetal health. Innovations in bio-potential, inertial, pressure, electrodermal, and acoustic sensors have enabled more precise detection and management of pregnancy-related conditions. The integration of multimodal sensors, combining various physiological signals, paves the way for personalized and cost-effective healthcare solutions by providing comprehensive health assessments.

The algorithms discussed in this review are designed for real-time and long-term monitoring using wearable sensors. Preprocessing techniques for raw data have become highly sophisticated, with some sensors incorporating filtering and noise-reduction features. While threshold-based algorithms are still in use, AI-based algorithms are increasingly replacing them. Many studies highlight the importance of machine learning in signal processing and classification, with numerous algorithms achieving over 90% accuracy in real-world applications. Moreover, AI is now being utilized beyond signal processing, offering early detection, intelligent diagnosis, and lifelong health management, marking a significant advancement in maternal and fetal monitoring.

Wearable sensor technology has the potential to significantly reduce healthcare disparities, especially in resource-limited regions and among vulnerable populations. By improving access to health information and preventive services, this technology can help address healthcare inequality. Firstly, wearable devices enable remote diagnostics, reducing the reliance on hospitals for pregnant women and other patients in underserved areas, thus mitigating challenges in accessing healthcare. Secondly, wearable sensors facilitate early diagnosis and timely intervention without the need to wait for symptoms to worsen. For example, early treatment of fetal distress is often more effective, which in turn helps to reduce healthcare expenditures. These developments suggest that future wearable devices will offer comprehensive health assessments, making personalized and affordable healthcare solutions a reality. However, the economic feasibility of these technologies remains a crucial factor. While the benefits of early diagnosis and continuous monitoring are clear, the cost of developing and scaling such advanced sensors could limit their accessibility, particularly in resource-constrained settings. Comprehensive cost-effectiveness analyses are needed to evaluate the scalability of wearable sensor technology, ensuring it remains affordable and effective in diverse healthcare environments. It is important to note that current research on wearable sensors for pregnant women heavily relies on smart devices, which may not be applicable in extremely poor areas. Therefore, it remains uncertain whether wearable sensors have truly reduced healthcare inequalities, and whether their economic feasibility requires further investigation.

On the other hand, although intelligent wearable sensors and associated data-processing methods offer transformative potential in pregnancy monitoring, they also face significant technical challenges that need to be addressed to fully realize their benefits. Long-term wearability, comfort, privacy, and usability are major concerns for users, posing design challenges for developers. In terms of applications, several issues need to be resolved, including technical compatibility and integration, device durability, battery life, cost, and compliance with international standards and regulations. In terms of data processing, key challenges include ensuring stable data transmission and storage, as well as protecting data privacy. Wearable sensors will generate vast amounts of personal health data, and future developments must focus on data security by employing advanced encryption technologies to prevent the leakage of sensitive maternal health data. Privacy-protection regulations, such as GDPR and HIPAA, could further regulate the collection and use of such data. Additionally, related regulations and standards still require further refinement to keep up with these emerging technologies. For the development, although new materials and designs have improved sensor sensitivity and comfort, future research must prioritize reducing power consumption, minimizing interference from movement, and enhancing overall integration to achieve seamless real-time monitoring. These areas represent important opportunities for further research and development.

In conclusion, while further research is necessary to ensure long-term safety, reliability, and cost-effectiveness, wearable sensors hold great promise for transforming maternal and fetal health outcomes worldwide.

## Figures and Tables

**Figure 1 sensors-24-06426-f001:**
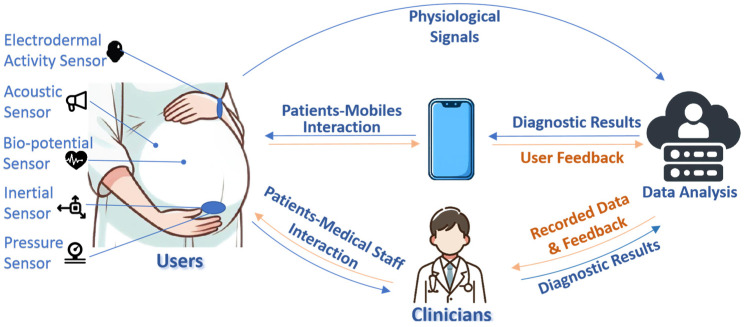
The pregnancy-monitoring system with smart wearable sensors.

**Figure 2 sensors-24-06426-f002:**
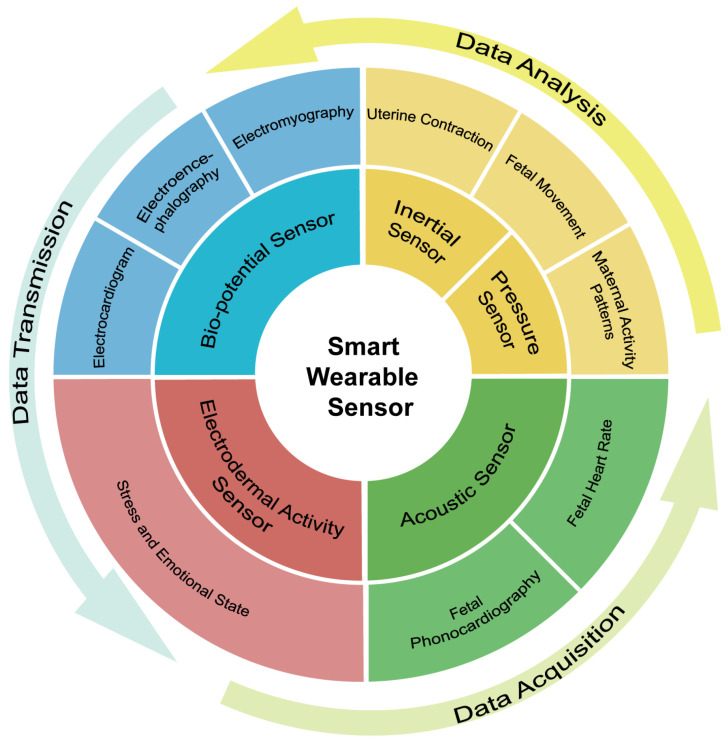
Classification and overview of the wearable sensors in pregnancy monitoring.

**Figure 3 sensors-24-06426-f003:**
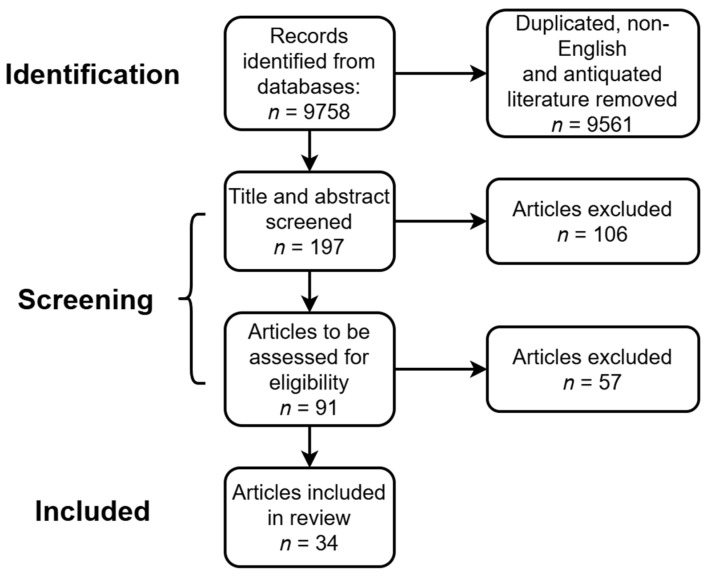
Flow diagram of screening strategy.

**Figure 4 sensors-24-06426-f004:**
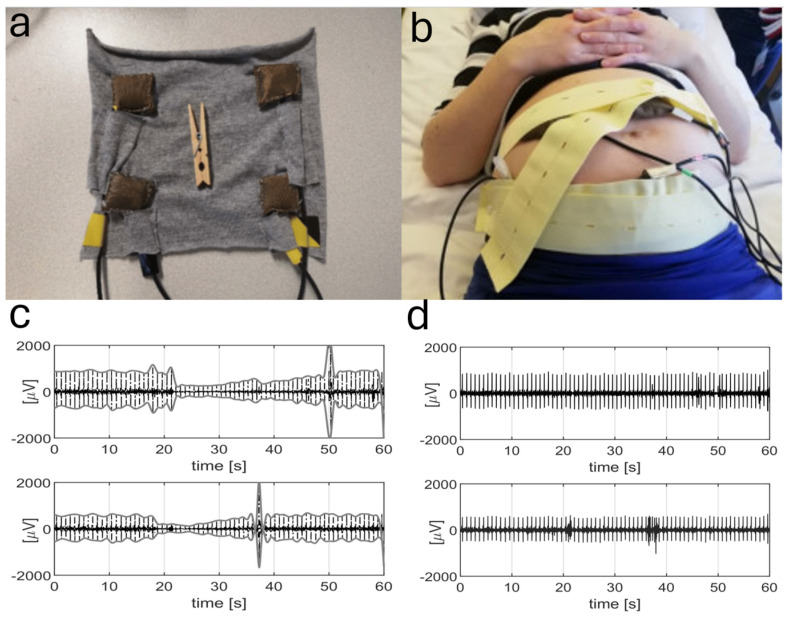
fECG monitoring system: (**a**) A close-up of four textile electrodes designed for fECG signal detection; (**b**) the non-invasive placement of electrodes using elastic straps around the expectant maternal abdomen; (**c**) raw signals (depicted in black) subjected to amplitude modulation, accompanied by their upper and lower envelopes (depicted in grey); (**d**) signals after the compensation of modulation. Reproduced with permission from [14]. Copyright © 2021, MDPI.

**Figure 5 sensors-24-06426-f005:**
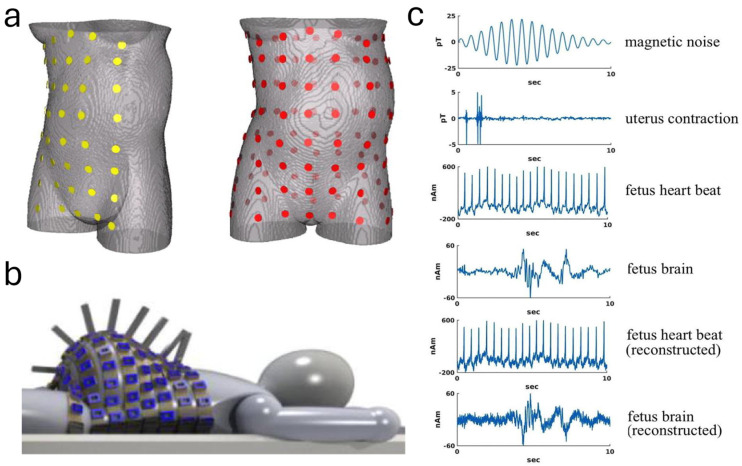
(**a**) A torso with two sensor-array configurations. The left part is a partial-coverage array (8 × 5 array, 40 channels). And the right part is a full-coverage array (8 × 16 array, 128 channels); (**b**) design of the FM scanner, which features a full-coverage sensor array made up of OPM sensors embedded in lightweight, flexible belts that are placed around the mother’s body. These sensor arrays are designed for detailed electrical activity mapping; (**c**) the signals of FHR and brain activity with the presence of external noise and internal interference. Reproduced with permission from [23]. Copyright © 2017, International Federation of Clinical Neurophysiology. Published by Elsevier Ireland Ltd.

**Figure 6 sensors-24-06426-f006:**
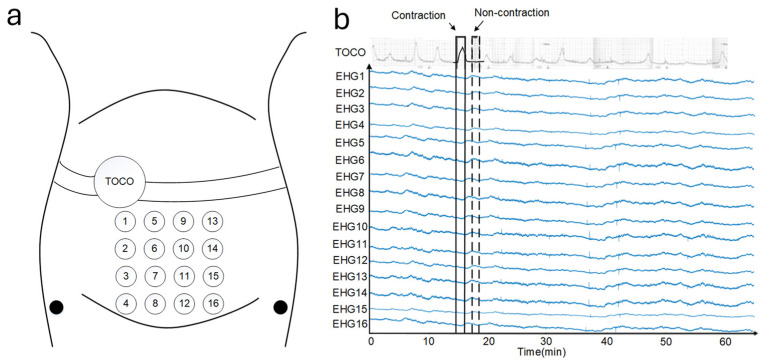
(**a**) Arrangement of EHG sensor electrodes; (**b**) sample of 16-channel EHG and TOCO recordings from the Icelandic 16-electrode EHG dataset. Reproduced with permission from [30]. Copyright © 2019 The Authors. Published by Elsevier Ltd.

**Figure 7 sensors-24-06426-f007:**
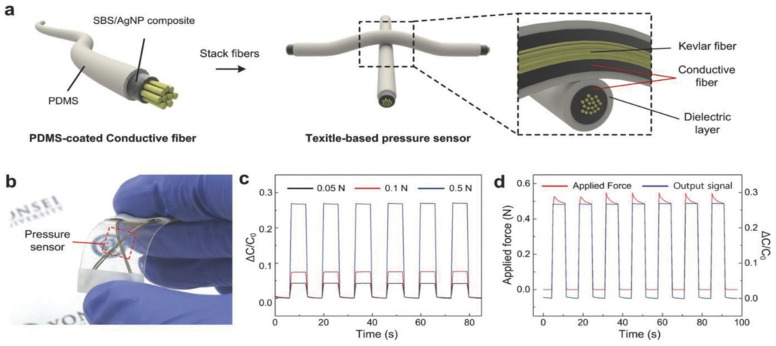
(**a**) Schematic structure of the textile-based pressure sensor; (**b**) photograph of the textile-based pressure sensor; (**c**) capacitive response under varying loads; (**d**) applied force and response curves under consistent loads. Reproduced with permission from [35]. Copyright © 2015 WILEY-VCH Verlag GmbH & Co. KGaA, Weinheim.

**Figure 8 sensors-24-06426-f008:**
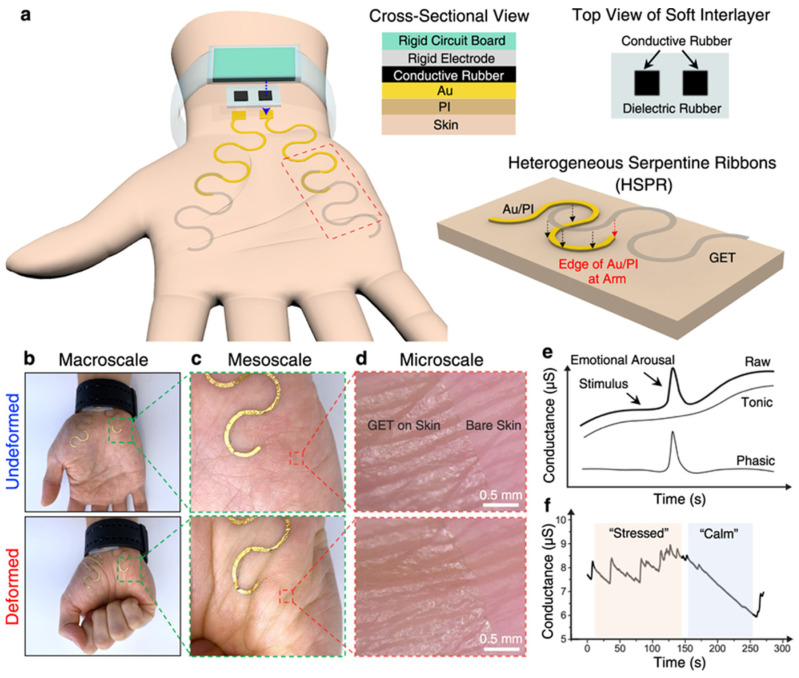
Multiscale views and signal response of a graphene-based EDA sensor. (**a**) Schematic of the sensor applied to the palm with key components labeled; (**b**–**d**) the sensor at macroscale, mesoscale, and microscale during application; (**e**) conductance response graph related to emotional arousal; (**f**) different conductance levels, indicating “stressed” and “calm” states. Reproduced with permission from [39]. Copyright © 2022, The Author(s). Published by Springer Nature.

**Figure 9 sensors-24-06426-f009:**
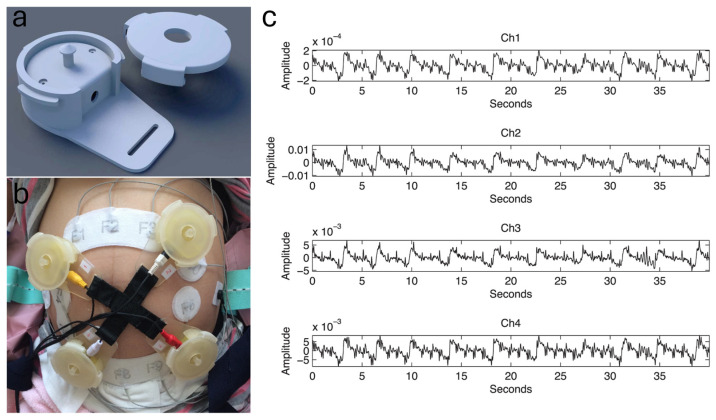
(**a**) Illustration of one of the four-channel fetal phonocardiogram sensor designs, including a top cover and a container; (**b**) actual application of the sensor on a pregnant woman’s abdomen, detailing the actual layout of fECG electrodes and proposed acoustic sensors; (**c**) four-channel raw fetal phonocardiogram recorded by the proposed sensors. Channels 1–4 display maternal respiratory movements (appearing as low-frequency baseline fluctuations), as well as maternal and fetal heart activity (visible as high-frequency fluctuations). Reproduced with permission from [49]. Copyright © 2018, The Author(s). Published by Springer Nature.

**Figure 10 sensors-24-06426-f010:**
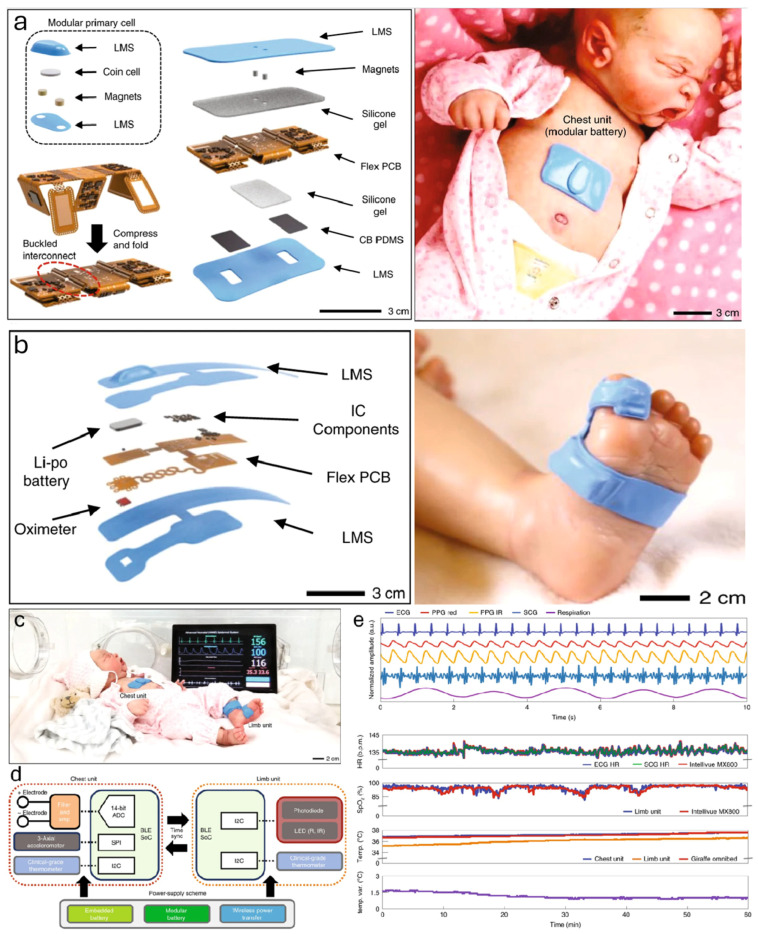
(**a**) A wearable device at the chest of a baby, with a flexible primary battery, featuring a soft enclosure and magnetic connectors for secure and gentle skin interfacing; (**b**) the limb unit’s design, detailed in schematic and exploded views, which is equipped to monitor photoplethysmography (PPG), blood oxygen saturation (SpO_2_), and skin temperature, powered by a lithium polymer battery; (**c**,**d**) illustrative diagram of two coordinated devices: a chest unit with systems for ECG, motion detection, temperature measurement, and Bluetooth transmission, as well as a limb unit with pulse oximetry, temperature sensing, and Bluetooth capabilities; (**e**) waveform data from a neonate (29 weeks gestational age), including ECG and SCG from a chest device and PPG from a limb device using red and infrared light, with respiration rates derived from chest readings. A comparison of vital signs and temperature data from these units with standard clinical tools confirms their accuracy. Reproduced with permission from [53]. Copyright © 2020, The Author(s), under exclusive licence to Springer Nature America, Inc.

**Figure 11 sensors-24-06426-f011:**
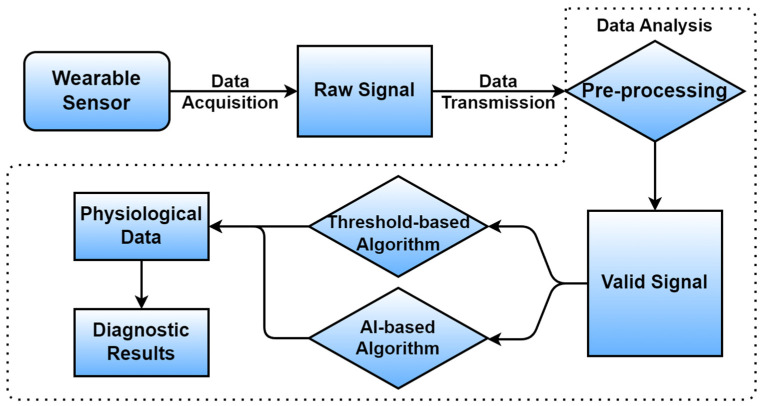
General data-processing flow in smart pregnancy monitoring.

**Figure 12 sensors-24-06426-f012:**
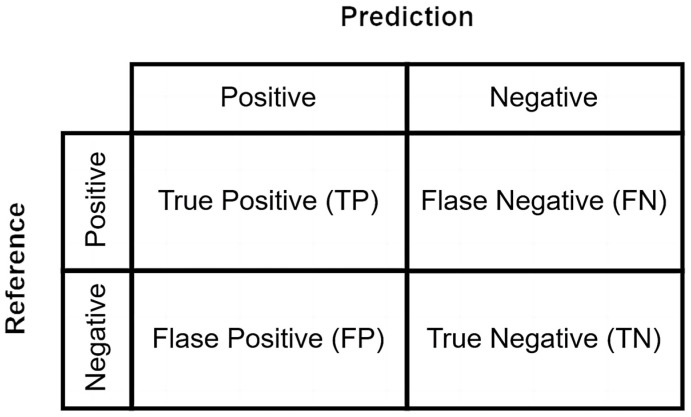
Confusion matrix of AI-based algorithms.

**Table 1 sensors-24-06426-t001:** Inclusion and exclusion strategy of this review.

Inclusion Strategy	Exclusion Strategy
Articles published in peer-reviewed venues	Articles not written in English
Articles published since 2000	Articles not including wearable sensors or pregnancy monitoring
Articles contained a certain combination of words, e.g., (wearable sensor OR wearable) AND ((maternal OR pregnant) OR (artificial intelligence OR AI))	Articles about qualitative interview surveys
	Articles excessively antiquated

**Table 2 sensors-24-06426-t002:** Description of recent research based on acquired signals, transmission methods, and algorithms. Note: The sample in [63] is from publicly available datasets, and the studies in [64,65] use multiple machine-learning methods, with only some listed.

Data	Sensor	Transmission Method	Pre-Processing	Feature Extraction	Algorithm	Performance	Ref.
**FM**	Accelerator	-	CWT	TFWD features	LightGBM Model	Accuracy: 94.06%	[63]
**FM**	Accelerator	PowerLab System	CWT and IIR	ICA and its first four discrete wavelets	Linear Regression	Accuracy: 87.6% to 95.8%	[64]
**FM**	Accelerator	Data-Acquisition (DAQ) System	IIR	Intersection of detection and sensation	Multiple Machine Learning Models	Accuracy: neural network and SVM: 90%; others: below 90%	[66]
**FM**	Accelerator	Bluetooth	Kalman	K-SVD dictionary learning	Orthogonal Matching Pursuit Algorithm	Positive prediction value: 89.74%.	[67]
**FM**	Ultrasound and Accelerator	Wired Protocol	CWT and Hampel	Statistical, morphological, and wavelet features	Ensemble Learning Model and Decision Tree	Accuracy: ensemble methods 93%; decision tree below 80%	[65]
**FHR& ECG**	Biopotential and Acoustic Sensor	Bluetooth	FIR	QRS detection	Peak Point Detection	−0.3 ppm from CTG FHR & 0.28 ppm from CTG MHR	[62]
**FHR**	Ultrasound Sensor	Wired Protocol	CWT	Systole extraction	Local Maximum Detection	Accuracy: 95.03%	[68]
**UC**	Force Sensor	Bluetooth	CWT and IIR	Features based on experience	Neural Network	Accuracy: 96%	[69]
**UC&MHR**	Biopotential and Acoustic Sensor	Bluetooth and WIFI	FIR and IIR	-	K-Means Clustering	Sensitivity: 94%; False Discovery Rates: 31.1%	[70]
**ECG**	Biopotential Sensor	Bluetooth	FIR and IIR	QRS detection	ADT and ICA	Average ACC: 97.40%; F1 score: 98.66%	[71]

Abbreviations: **FM**: Fetal Movement; **FHR**: Fetal Heart Rate; **ECG**: Electrocardiogram; **UC**: Uterine Contraction; **MHR**: Maternal Heart Rate; **DAQ**: Data Acquisition; **ADT**: Adaptive Dual Threshold; **FIR**: Finite Impulse Response; **IIR**: Infinite Impulse Response; **CWT**: Continuous Wavelet Transformation; **DWT**: Discreate Wavelet Transformation.

**Table 3 sensors-24-06426-t003:** Performance comparison of different communication protocols.

Transmission Method	Power Consumption	Security	Latency	Bandwidth	Communication Range	Ref.
**Bluetooth**	Very Low	Normal	Milliseconds to hundreds of milliseconds	1 Mbps or more	10–100 m	[65,67,69,70,71]
**WIFI**	High	Strong	tens of milliseconds	Tens of Mbps to hundreds of Mbps	50–100 m	[70]
**Wired Protocol**	Low	Very Strong	Several milliseconds	Depends on system	Depends on system	[65,68]
**LoRa**	Very Low	Weak	Several Seconds	0.3–50 kbps	Several km	[73,74]
**NB-IoT**	Low	Strong	1–10 s	About 250 kbps	Several km	[72]

## Data Availability

Not applicable.

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
