# Peer review of "Wearable Sensors, Data Processing, and Artificial Intelligence in Pregnancy Monitoring: A Review"

_sensors, 2024, doi:10.3390/s24196426_

Round 1

Reviewer 1 Report

Comments and Suggestions for Authors

In this review entitled “Wearable Sensors, Data Processing and Artificial Intelligence in Pregnancy Monitoring: A Review” by Liu et al., the applications of wearable sensors in pregnancy monitoring are introduced and discussed. In this paper, there some issues should be addressed by the authors.

1.What happens to Figures 1, 2, 3, 11, and 12? Why there appear in the manuscript for so many times? This should be revised by the authors.

2.Some explanations for Table 1 are suggested.

3.Copyright permission is missing for all the Figures, which should be added by the authors.

4.Some recent published related to the wearable should cited in this review. For example,https://doi.org/10.1021/acsami.4c07732

Comments on the Quality of English Language

Good

Author Response

Comments 1: What happens to Figures 1, 2, 3, 11, and 12? Why there appear in the manuscript for so many times? This should be revised by the authors.

Response 1: Thanks for the heads up. I have checked and corrected the corresponding image and the caption.

Comments 2: Some explanations for Table 1 are suggested.

Response 2: Thank you for the heads up. The inclusion and exclusion policies in the table have been explained in the main text in the revision.(line 132-139)

Comments 3: Copyright permission is missing for all the Figures, which should be added by the authors.

Response 3: We appreciate the feedback. Copyright permission for all the figures has been added as requested.

Comments 4: Some recent published related to the wearable should cited in this review. For example,https://doi.org/10.1021/acsami.4c07732

Response 4: Thank you for the suggestion. We have incorporated the recommended citation (https://doi.org/10.1021/acsami.4c07732) into Section 3 (line 435) to ensure it includes the most recent relevant literature on wearable sensors. And some citations such as doi.org/10.1109/TII.2024.3424517, doi.org/10.3390/s23218885, doi.org/10.1109/IWCMC48107.2020.9148359 and doi.org/10.1109/ACCESS.2021.3082006 (Table 3, line 710) are added in Section 5.

Reviewer 2 Report

Comments and Suggestions for Authors

The author provides a comprehensive overview of wearable sensors and AI technologies in pregnancy monitoring. It outlines the significance of these technologies in improving home-based health monitoring. The manuscript can be considered for publication after addressing the following concerns.

1.      The article should provide a more detailed comparison between traditional monitoring methods and the new wearable sensor technologies.

2.      It is suggested to elaborate on the data privacy and security when handling sensitive health data collected by wearable sensors.

3.      Are there any long-term studies on the safety and reliability of wearing these sensors throughout pregnancy?

4.      The economic implications and cost-effectiveness of implementing wearable sensor technology in prenatal care can be included.

5.      More information should be provided on how the data from these sensors can be integrated into electronic health records (EHRs) and the challenges associated with such integration.

6.      Can the authors comment on the potential impact of these wearable sensor technologies on reducing healthcare disparities.

7.      It would be useful to include discussion on future expectation that discusses upcoming technologies or directions in wearable sensor research for pregnancy monitoring.

8.      Ensure that all cited studies and technologies are up-to-date and relevant. Recent references with the latest 3 years on cutting-edge research or emerging technologies should be included.

Comments on the Quality of English Language

The writing requires further polishing.

Author Response

Comments 1: The article should provide a more detailed comparison between traditional monitoring methods and the new wearable sensor technologies.

Response 1: Thank you for the reminder. We have further adjusted the text to enhance this comparison and clarify the advantages of wearable sensors in pregnancy health monitoring.(line 1022-1028; 1038-1048)

Comments 2: It is suggested to elaborate on the data privacy and security when handling sensitive health data collected by wearable sensors.

Response 2: Thank you for pointing this out.Section 1, a detailed description of the challenges and limitations of wearables and AI applications, including data privacy and security issues, has been added (line 101).A related description has been added to Section 5.1, Data Transmission.A separate paragraph has been added to Section 5.4 to analyse the data privacy and security issues and to give a possible future solution (line 927). Section 5.5, a paragraph has been added to describe the risks and potential solutions, including data privacy issues (line 999). Section 5.5, a paragraph has been added describing risks and potential solutions, including data privacy issues (line 999).

Comments 3: Are there any long-term studies on the safety and reliability of wearing these sensors throughout pregnancy?

Response 3: Thank you for the insightful comment. Currently, there are limited long-term studies specifically focused on the safety and reliability of wearable sensors throughout pregnancy. In the manuscript, we have highlighted this gap in the literature and emphasized the need for future research to address the long-term wearability, potential risks, and performance of these sensors over extended periods of use during pregnancy.(line 1061)

Comments 4: The economic implications and cost-effectiveness of implementing wearable sensor technology in prenatal care can be included.

Response 4: Thank you for raising this important point. We have now included a discussion on the economic implications and cost-effectiveness of implementing wearable sensor technology in prenatal care. While the benefits of improved pregnancy monitoring and early diagnosis are clear, challenges remain in making these technologies cost-effective and accessible, especially in low-resource settings. We acknowledge the need for further research into the cost analysis and scalability of these solutions to ensure their widespread adoption. (line 1048-1057)

Comments 5: More information should be provided on how the data from these sensors can be integrated into electronic health records (EHRs) and the challenges associated with such integration.

Response 5: Thank you for the reminder. We have added an overall description at the end of Section 5.1. Describes the general process of data integration and possible challenges. (line704;734;980)

Comments 6: Can the authors comment on the potential impact of these wearable sensor technologies on reducing healthcare disparities.

Response 6: Section 6 adds a discussion and outlook in this regard. The authors believe that wearable sensors can be effective in reducing healthcare inequalities, but not enough research has been done to analyse the economic costs. Economic feasibility may be an issue.(line 1038-1048)

Comments 7: It would be useful to include discussion on future expectation that discusses upcoming technologies or directions in wearable sensor research for pregnancy monitoring.

Response 7: Thank you for the suggestions. We have added a discussion on future expectations and opportunities in wearable sensor research for pregnancy monitoring, focusing on emerging technologies such as AI integration, flexible materials, and IoT-based systems. These areas represent key directions for advancing the field and improving long-term reliability and accessibility. (line 1062-1075)

Comments 8: Ensure that all cited studies and technologies are up-to-date and relevant. Recent references with the latest 3 years on cutting-edge research or emerging technologies should be included.

Response 8: We confirm that all the key citations are from the past five years, with the majority being from the last three years to ensure the review reflects the most recent advancements in wearable sensor technologies. We also have incorporated the most recent work like doi.org/10.1021/acsami.4c07732 into Section 3 (line435). And some citations such as doi.org/10.1109/TII.2024.3424517 and doi.org/10.1109/ACCESS.2021.3082006 are added in Section 5 (table 3, line710)to strengthen our review.

Reviewer 3 Report

Comments and Suggestions for Authors

1.      In the review methodology, the flow diagram in Fig. 3 seems redundant.

2.      In the literature of review of ECG, detailed pros of the reported works are addressed while their corresponding limitations are missing.

3.      I would suggest adding how the AI as a promising tool has been integrated in pregnancy monitoring utilizing EHG sensor electrodes, as claimed in line 296 of the manuscript.

4.      As this proposed review work focuses on the application of various sensors in pregnancy monitoring, the inertial sensors and pressure sensors are somehow to be correlated with the pregnancy health monitoring with some published research works, which seems missing. So, section 3.2. Inertial Sensor & Pressure Sensor must be revised in the context of pregnancy monitoring issues.

5.      Same concern arises for 3.3. Electrodermal Activity Sensor, where the authors have described measuring features of sensors without any exemplary issues in pregnancy health monitoring by those electrodermal activity sensors.

6.      I would suggest explaining the working principles behind the threshold-based algorithms more detail. Also, I expect the major insights for the poor performance of threshold-based algorithms.

7.      Finally, to highlight the role of IoT-based wearables in pregnancy health monitoring, the author should include some potential review works showcasing the integration of technology in healthcare, emphasizing the importance of innovation in improving maternal and fetal health.

Author Response

Comments 1: In the review methodology, the flow diagram in Fig. 3 seems redundant.

Response 1: Thank you for pointing out. This paper is a literature search based on the PRISMA-P methodology (10.1186/2046-4053-4-1), Fig.3 is mainly to show the standardisation process, and after discussion among the authors, it is considered better to keep Fig.3. If you have further comments, please don’t hesitate to let us know.

Comments 2: In the literature of review of ECG, detailed pros of the reported works are addressed while their corresponding limitations are missing.

Response 2: Thank you for the observation. We have now included a discussion of the corresponding limitations of the reported works in the ECG literature review. The section has been revised to provide a more balanced analysis, highlighting both the advantages and the limitations of the studies mentioned. (line 170;183;228;239;244;248, section 3.1.1)

Comments 3: I would suggest adding how the AI as a promising tool has been integrated in pregnancy monitoring utilizing EHG sensor electrodes, as claimed in line 296 of the manuscript.

Response 3: We appreciate the reviewer’s suggestion. We have expanded the discussion in the manuscript to detail how AI has been integrated with EHG sensor electrodes in pregnancy monitoring, as referenced in line 296. This addition provides a comprehensive overview of the role of AI in enhancing the accuracy and efficiency of EHG-based monitoring systems.(line 314;317)

Comments 4: As this proposed review work focuses on the application of various sensors in pregnancy monitoring, the inertial sensors and pressure sensors are somehow to be correlated with the pregnancy health monitoring with some published research works, which seems missing. So, section 3.2. Inertial Sensor & Pressure Sensor must be revised in the context of pregnancy monitoring issues.

Response 4: Thank you for the comments. We have revised sections 3.2 and 3.3 to better correlate inertial, pressure, and electrodermal activity sensors with pregnancy health monitoring. Examples have been added to show how these sensors can track fetal movements, uterine contractions, and maternal stress, supporting their application in pregnancy monitoring.(line 353;359;364;369;379;385)

Comments 5: Same concern arises for 3.3. Electrodermal Activity Sensor, where the authors have described measuring features of sensors without any exemplary issues in pregnancy health monitoring by those electrodermal activity sensors.

Response 5: Thank you for the comments. We have revised sections 3.2 and 3.3 to better correlate inertial, pressure, and electrodermal activity sensors with pregnancy health monitoring. Examples have been added to show how these sensors can track fetal movements, uterine contractions, and maternal stress, supporting their application in pregnancy monitoring.(line 408;414;435;462)

Comments 6: I would suggest explaining the working principles behind the threshold-based algorithms more detail. Also, I expect the major insights for the poor performance of threshold-based algorithms.

Response 6: Thank you for the reminder. In the Threshold-based Algorithm section, an explanation of the basics of the threshold algorithm and possible reasons for poor performance have been added at the Introduction to the Algorithm. (line815)

Comments 7: Finally, to highlight the role of IoT-based wearables in pregnancy health monitoring, the author should include some potential review works showcasing the integration of technology in healthcare, emphasizing the importance of innovation in improving maternal and fetal health.

Response 7: Thank you for the suggestion. We have added a discussion highlighting the role of IoT-based wearables in pregnancy health monitoring. We also have incorporated the most recent work like doi.org/10.1021/acsami.4c07732 into Section 3 (line435) that showcase the integration of IoT technology in healthcare, emphasizing how innovations in this field can improve maternal and fetal health outcomes.

Reviewer 4 Report

Comments and Suggestions for Authors

The paper is interesting and addresses a complex problem of interest to everyone. In general, it seems to me to be well structured and addresses important aspects for the development of monitoring platforms. Here are some observations that in my opinion could significantly help to improve the paper.

- The introduction could include more information on current limitations or challenges in the use of wearable sensors and AI.

- In Section 3, it would be very useful to include specific examples of how these algorithms have been tested or validated in real-world scenarios, with data obtained from wearable sensors. In this same section, a table with the different algorithms used with descriptions and important characteristics would also be useful to be able to appreciate them in a simple way.

- On the other hand, when talking about monitoring platforms, the architectures used, including hardware, software, protocols, etc., are regularly mentioned. None of the ones investigated is shown or mentioned regularly. A comparative table of the different architectures should be considered to determine which technology is being used to develop these monitoring platforms.

- A more detailed comparison table should be added between the different communication protocols including security and energy efficiency aspects for example.

-- Devices with sensors have mostly limited computing power, so the discussion of data processing could be extended, for example there are some QoS (Quality of Service) metrics in some software architectures to improve the energy consumption of the device and not transmit all data completely, for example a filter type QoS could avoid transmitting all data and only let “pass” to the medium the data according to the QoS.

- Similarly, with respect to data processing in section 5, mention in a general way data acquisition, transmission through certain protocols and the “data analysis module”, data are stored for later analysis, analyzed either in real-time or data are downloaded for further analysis. What database technology is used for these platforms, this of course is part of the system architecture, and it is always important to know it.

- The risks and limitations of using AI in medical diagnostics should be more detailed, including possible errors and how these can be handled in real life.

- One of the problems I have always seen in the developments is that of economic feasibility, no cost analysis of these technologies is presented, smart phones are mentioned but they are not accessible to many people. How feasible is it to develop this technology in terms of cost? The benefit of good pregnancy monitoring is clearly understandable and early diagnosis can save many lives, but how convenient is it?

- It would be interesting to know some areas of opportunity detected in the work, i.e. what lines of action could be interesting to address.

- I hope that these observations are pertinent and do not go beyond the scope of your research, and if you have other ideas to improve the writing derived from these, you are free to make them in the interest of improving the paper.

Author Response

Comments 1: The introduction could include more information on current limitations or challenges in the use of wearable sensors and AI.

Response 1: Thanks for the heads up. In the first part, the authors added a detailed description of the challenges and limitations facing wearables and AI applications. (line101)

Comments 2: In Section 3, it would be very useful to include specific examples of how these algorithms have been tested or validated in real-world scenarios, with data obtained from wearable sensors. In this same section, a table with the different algorithms used with descriptions and important characteristics would also be useful to be able to appreciate them in a simple way.

Response 2: Thanks for the heads up and suggestions. Since the third part is mainly narrowed down to the hardware part, the algorithms will not be analysed in detail. The analysis of the performance and efficiency of different algorithms is mainly concentrated in Section 5 (pre-processing: 5.2; data analysis methods: 5.3). If you have further comments, please don’t hesitate to let us know.

Comments 3: On the other hand, when talking about monitoring platforms, the architectures used, including hardware, software, protocols, etc., are regularly mentioned. None of the ones investigated is shown or mentioned regularly. A comparative table of the different architectures should be considered to determine which technology is being used to develop these monitoring platforms.

Response 3: Thanks for the heads up. The authors have categorized the systems in table2 according to the different hardware, transmission protocols and algorithms used. A more detailed description is given in the relevant part of the text. (Lines: 755; 764; 836; 912) Since the data processing part mainly focuses on the discussion of algorithmic features, and the analysis of the architecture of the software system is beyond the author's expertise, and is prone to omissions or errors, the architecture of the system has not been singled out for comparison, and only an example of a system with a complete wearable device has been illustrated. If there are any problems, please feel free to point them out.

Comments 4: A more detailed comparison table should be added between the different communication protocols including security and energy efficiency aspects for example.

Response 4: Thanks for the heads up. The authors have re-queried the communication protocols that have been applied to health monitoring related areas. The authors have added a relevant comparison table (Table 3, line 710) in the data transmission section, which analyses the differences between different communication protocols, including communication distance, energy consumption, bandwidth, etc. and explained in the text.

Comments 5: Devices with sensors have mostly limited computing power, so the discussion of data processing could be extended, for example there are some QoS (Quality of Service) metrics in some software architectures to improve the energy consumption of the device and not transmit all data completely, for example a filter type QoS could avoid transmitting all data and only let “pass” to the medium the data according to the QoS.

Response 5: Thanks for the heads up. The authors expand on the application of QoS metrics to wearables and smart healthcare in the Pre-processing section. (line 717-730)

Comments 6: Similarly, with respect to data processing in section 5, mention in a general way data acquisition, transmission through certain protocols and the “data analysis module”, data are stored for later analysis, analyzed either in real-time or data are downloaded for further analysis. What database technology is used for these platforms, this of course is part of the system architecture, and it is always important to know it.

Response 6: Thanks for the heads up. A brief discussion has been added to Section 5.4 (lines 952-958). Unfortunately there is no specific literature analyzing and summarizing the databases used in wearable systems. Database technology is beyond the expertise of the authors and there may be omissions or errors, so feel free to point them out if there are any.

Comments 7: The risks and limitations of using AI in medical diagnostics should be more detailed, including possible errors and how these can be handled in real life.

Response 7: A paragraph has been added at Section 5.5 AI describing the risks and potential solutions. Includes: dependence of AI models on datasets; opacity of decision-making process; data privacy issues. (line 999-1014)

Comments 8: One of the problems I have always seen in the developments is that of economic feasibility, no cost analysis of these technologies is presented, smart phones are mentioned but they are not accessible to many people. How feasible is it to develop this technology in terms of cost? The benefit of good pregnancy monitoring is clearly understandable and early diagnosis can save many lives, but how convenient is it?

Response 8: Thank you for raising this important point. We have now included a discussion on the economic implications and cost-effectiveness of implementing wearable sensor technology in prenatal care. While the benefits of improved pregnancy monitoring and early diagnosis are clear, challenges remain in making these technologies cost-effective and accessible, especially in low-resource settings. We acknowledge the need for further research into the cost analysis and scalability of these solutions to ensure their widespread adoption.(line 978)

Comments 9: It would be interesting to know some areas of opportunity detected in the work, i.e. what lines of action could be interesting to address.

Response 9: Thank you for the reminder. In the Conclusion section, we address the potential opportunities in this field by analyzing the current challenges faced by wearable sensors. (line 1062-1068).